# Silicon-Mediated Interactions Between Plant Antagonists

**DOI:** 10.3390/plants14081204

**Published:** 2025-04-14

**Authors:** Marie-Emma Denarié, Uffe N. Nielsen, Susan E. Hartley, Scott N. Johnson

**Affiliations:** 1Hawkesbury Institute for the Environment, Western Sydney University, Locked Bag 1797, Penrith, NSW 2751, Australia; m.denarie@westernsydney.edu.au (M.-E.D.); u.nielsen@westernsydney.edu.au (U.N.N.); 2School of Biosciences, University of Sheffield, Sheffield S10 2TN, UK; s.hartley@sheffield.ac.uk

**Keywords:** silica, insect herbivores, nematodes, phytopathogens, phytohormones

## Abstract

The prolonged arms race between plants and their antagonists has resulted in the evolution of multiple plant defence mechanisms to combat attacks by pests and pathogens. Silicon (Si) accumulation occurs mainly in grasses and provides a physical barrier against antagonists. Biochemical pathways may also be involved in Si-mediated plant resistance, although the precise mode of action in this case is less clear. Most studies have focussed on Si-based effects against single attackers. In this review, we consider how Si-based plant resistance operates when simultaneously and/or sequentially attacked by insect herbivores, fungal phytopathogens, and plant parasitic nematodes and how the plant hormones jasmonic acid (JA) and salicylic acid (SA) are involved. Si defence may mediate both intra- and interspecific competition and facilitation. Si has been found to impact plant-mediated interactions between insect herbivores within the same feeding guild and across different feeding guilds, with varying patterns of JA and SA. These results suggest that hormonal crosstalk may play a role in the Si-mediated effects, although this finding varied between studies. While some reports support the notion that JA is linked to Si responses, others indicate that Si supplementation reduces JA production. In terms of phytopathogens, SA has not been found to be involved in Si-mediated defences. Improving our understanding of Si-mediated plant defence could be beneficial for sustainable agriculture under future climates.

## 1. Introduction

As the global population continues to grow, securing adequate levels of food production is becoming increasingly challenging [1]. At the same time, yields of staple crops such as rice (*Oryza sativa*), maize (*Zea mays*), wheat (*Triticum aestivum*), and potato (*Solanum tuberosum*) are threatened by pests and diseases: 26–40% of crops are lost to weeds, insects, nematodes, and pathogens [2,3], losses that are predicted to increase under climate change [4]. Nevertheless, plants can defend themselves against many harmful pathogens and invertebrates through a combination of pre-existing physical and chemical barriers and inducible defence responses that are activated after attack. Phytohormones such as salicylic acid (SA), jasmonic acid (JA), and ethylene (ET) play a vital role in the signalling pathways conferring these defences [5,6]. On the one hand, the SA pathway initiates defence against (hemi)biotrophic fungi, oomycetes, viruses, and piercing–sucking phloem feeders through the induction of systemic acquired resistance (SAR) [7]. On the other hand, JA and ET are involved in induced systemic resistance (ISR), which is triggered by chewing herbivores, bacteria, nematodes, and necrotrophs [6,8].

One defence mechanism particularly more prevalent in grasses than in other plant taxa is silicon (Si) accumulation [9,10]. Grasses, including the most important food crops globally, wheat and rice, are known to show increased resistance to a range of abiotic [11,12] and biotic stresses [13,14] after Si absorption. The uptake, transport, and deposition of Si in plants has been characterised well: it enters plant roots as silicic acid [Si(OH)_4_], is moved through the plant via transportation in the transpiration stream and active transport [15], and is deposited as insoluble phytoliths in a range of plant tissues [16]. Although Si is abundantly present in the Earth’s crust, plant available Si(OH)_4_ can become deficient in agricultural soils [17]. Hence, the supplementation of Si to plants in an agricultural context is a growing area of interest. Si amendment in laboratory and field conditions decreases plant susceptibility to various antagonists through its impact on defence responses against herbivorous insects, pathogenic fungi, and plant parasitic nematodes (PPNs) [8,18,19,20,21].

To our knowledge, most previous studies have investigated Si-mediated defences on single attackers, with the interplay between Si-mediated defences against multiple attackers remaining largely underexplored. In this mini review, we first discuss Si-based defences against different antagonist taxa, focussing on insect herbivores, plant parasitic nematodes, and fungal plant pathogens. Second, we consider the potential for plant-mediated interactions in the context of phytohormones. Finally, we review the limited evidence for Si-based interactions between plant antagonists and highlight the knowledge gaps in this field. Our objective is not to provide an exhaustive review of Si defences against these three antagonist taxa, which has been performed elsewhere [8,20,22,23], but to identify the salient impacts of Si on defensive responses (i.e., phytohormonal signalling) and the potential consequences for plant-mediated interactions between attackers.

## 2. The Role of Plant Silicon in Response to Biotic Stresses

### 2.1. Insect Herbivores

The effects of Si uptake on plant defence against various antagonists have been widely reported, especially against herbivorous insects. Examples of pests that are known to be affected by plant Si are brown plant hopper (*Nilaparvata lugens*), stem borer (*Chilo polychrysus*), green leaf hopper (*Cicadella viridis*), white backed plant hopper (*Sogatella furcifera*), cotton bollworm (*Helicoverpa armigera*), native budworm (*Helicoverpa punctigera*), locusts (*Schistocerca gregaria*), African armyworm (*Spodoptera exempta*), and non-insect pests like spider mites (Tetranychidae) [24,25,26,27,28,29,30,31,32,33,34]. Si is thought to increase plant resistance to insects through physical as well as biochemical measures. However, it is most likely that enhanced resistance to herbivory is caused by Si deposition between and within plant cell walls, creating a physical barrier against attackers [35,36]. Insoluble silica (SiO_2_) is deposited as phytoliths in leaves, which increases the hardness of tissues [25,37], wearing herbivore mouthparts and reducing plant tissue digestibility [25,35,38]. Notably, there are studies that question whether silica causes mandibular wear [39]. However, silicified trichomes can cause gut damage to larvae feeding off Si-amended plants [37]. While this mechanism is effective against chewing herbivores, the significance of Si against other feeding guilds, such as sap-feeding insect herbivores, is not entirely clear [40]. Due to their haustellate mouthparts, piercing–sucking insects can avoid silica barriers in leaves during feeding, so they may be less affected compared to chewing herbivores [23,34]. However, some piercing–sucking insects have been found to be affected by Si in host plants, albeit to a lesser extent [40]. In addition to this physical line of defence, there is evidence that Si is linked to phytochemical pathways. For instance, Si has been reported to induce the accumulation of numerous biochemicals, such as reactive oxygen species (ROS), antioxidant enzymes, and callose [21]. Furthermore, evidence shows that Si triggers systemic stress signals mediated by phytohormones such as the JA pathway [10,41,42,43].

### 2.2. Fungal Pathogens

Si has been shown to decrease plant host susceptibility to a wide range of fungal pathogens [13,44,45,46,47,48,49,50,51,52,53,54]. Amendment of soil with Si effectively suppresses various diseases in rice, such as leaf spot, blast, and sheath blight. Furthermore, reports of decreased susceptibility to powdery mildew have been made in wheat, barley (*Hordeum vulgare*), cucumber (*Cucumis sativus*), bitter gourd (*Momordica charantia*), grape (*Vitis vinifera*), and muskmelon (*Cucumis melo*) [25,55,56,57]. Besides soil applications, foliar spraying of Si has been found promising in crop protection as well [58,59]. The severity and occurrence of both air- and soilborne diseases can be reduced by Si amendment [60]. Reductions in susceptibility to these pathogens are manifested by reduced lesion size and number, decreased colony size, delayed incubation period, and restricted reproduction of fungi [55,60]. Initially, it was believed that fungal penetration of plants was merely physically hampered by deposited silica below the leaf cuticle [25,55,60]. However, later, it has been acknowledged that there may be other factors at play [61], although many of the proposed mechanisms remain hypothetical. Van Bockhaven et al. [5] proposed five hypotheses to explain Si-mediated broad-spectrum defence: (1) Si-induced priming (modifying intensity and/or timing of basal defence responses), (2) Si-induced hormone interactions, (3) targeted alterations in iron homeostasis, (4) Si-driven photorespiration, and (5) Si interaction with signalling components (Figure 1). For instance, it is speculated that Si-mediated defence responses are conferred through plant hormones such as SA, JA, and ET. Additionally, it is thought that Si may induce the expression of downstream defence-related genes and hence triggers defence-related enzymes such as peroxidase (POX), phenylalanine ammonia lyase (PAL), and polyphenol oxidase (PPO) [25,55].

The lifestyle of fungi, i.e., their host range, nutrient source, and infection strategy, appears to be a factor that influences their response to Si accumulation by plants. Si has been shown to have a stronger efficacy against biotrophic [62] and hemibiotrophic pathogens [45] compared to necrotrophs [63]. Additionally, the hormone signalling pathways that are induced upon infection differ: SA by (hemi)biotrophs and JA and ET by necrotrophs [20,64]. Nonetheless, there are instances where necrotrophs appeared to be affected by Si, reflecting ambiguity in this distinction [65] and in the plant responses to the different lifestyles [48,66]. Coskun et al. [67] introduce a unifying model in which they explain the range of beneficial effects that Si mediates against biotic and abiotic stresses. Their proposed apoplastic obstruction hypothesis argues that fungal and herbivore effectors are physically hampered through silica deposition in the apoplast of plant cells. Consequently, effectors cannot hinder plant defences, and thus plant susceptibility is reduced [67]. Nevertheless, the precise mechanisms through which Si confers enhanced plant resistance to fungal pathogens remain to be determined.

### 2.3. Plant Parasitic Nematodes

Similarly to insects and fungi, the amendment of plants with Si has been found to reduce the severity of nematode attacks [68,69,70,71,72,73]. For instance, the number of root galls and eggs of various root knot species (*Meloidogyne* spp.) was reduced in beetroot (*Beta vulgaris*), cucumber, coffee (*Coffea arabica*), and tomato (*Solanum lycopersicum*) [6,74,75,76,77]. The mechanisms by which Si provides protection against nematodes are often unclear but may have some commonalities with defence against insect herbivores and pathogens. The formation of a physical barrier in cell walls through Si deposition, for example, may prevent nematode stylets from penetrating host tissues [6,8]. In the context of the biochemical mechanisms, Si is believed to initiate the production of phenol-like compounds; increase levels of phytoalexins POX, PPO, and PAL; and activate pathogenesis-related (PR) genes against nematodes [6,8,76,78]. Several studies suggest that Si may confer heightened plant immunity to pests through mediating hormone signalling pathways and phytohormone levels, i.e., JA, SA, and ET [8]. Nonetheless, there has been a limited number of studies investigating this.

## 3. Plant-Mediated Interactions Between Antagonists

In natural environments, plants are frequently subjected to simultaneous infestations by antagonists. For example, while plant foliage can be under attack by herbivorous caterpillars, root-feeding nematodes may infect its belowground parts. Consequently, they can interact indirectly through shared hosts by causing alterations in plant morphology and chemical defences [79]. Although changes in morphology have been studied mostly in aerial parts of plants instead of roots, root exudates have been reported to modify interactions between antagonists in the rhizosphere. For instance, tobacco (*Nicotiana benthamiana*) plants infected with tobacco rattle virus (TRV) have been shown to alter root volatile emissions and cause a higher attraction of trichodorid nematodes. Notably, these root-feeding nematodes can also act as vectors for TRV, aiding transmission within crops [80]. Injured roots can cause the nutritional status of the rhizosphere to increase, thereby promoting fungal growth or attracting insects. Sometimes, the interaction is so strong that the insect is viewed as part of the disease complex, for example root-feeding larval coleopterans and pathogens such as *Fusarium* spp. that are associated with root rot diseases [81]. In this case, there is a facilitative relationship between the antagonists. In contrast, when resources such as nutrition are limited, competition can occur when attackers coexist spatiotemporally on the same plant host [82,83].

### 3.1. Facilitation

Interactions between plant antagonists can greatly influence their fitness and performance—for instance, among different insect herbivores. An example is that stem-boring moths (*Endoclita excrescens*) induce shoot growth in willow trees (*Salix* spp.), which subsequently facilitates feeding by specialist leaf beetles (*Plagiodera versicolora*) [84]. Herbivory activates defence signalling pathways in plants, which can be differentially influenced by the presence of multiple insects. Certain plant defence mechanisms are mediated by phytohormones, which are induced differently depending on the identity, sequence, and intensity of attack of the varying stressors [83]. JA plays a role in responses against chewing insects, while both SA and JA are involved in the regulation of defences against fluid-feeding insects [7]. There can be antagonistic crosstalk between these two hormones; for instance, activation of the SA pathway usually suppresses JA signals. Consequently, defence reactions from hosts are finetuned to be tailored against specific herbivores based on their feeding guild. This may lead to facilitation, in which the feeding of one herbivore increases the performance of subsequent herbivores feeding on the same plant as ineffective defence signalling pathways are induced [7,35,85].

There are different factors that can influence phytohormonal response during the presence of multiple herbivore attackers, for example the host’s susceptibility to a specific insect, even within the same feeding guild. In addition, plant ontogeny, chronological order, and duration of stress play an important role [83]. Induced defence responses can occur within the same host tissue compartments, i.e., within the root system belowground or on the leaves aboveground [64,81]. Furthermore, such plant-mediated effects do not only occur within the same class as with insects but also across different taxa. For instance, herbivorous insect feeding may affect plant resistance to root-feeding nematodes and vice versa. The performance of the generalist phloem-feeding aphid *Myzus persicae* was found to be greater on potato plants that had been pre-infected with the sedentary endoparasitic cyst nematode *Globodera pallida*. This facilitative interaction was attributed to systemic biochemical changes in the potato plants, more specifically, SA and JA [86]. Notably, there is limited knowledge of hormone signalling and systemic responses induced by nematodes in roots. It is known, however, that sedentary endoparasitic nematodes induce SA, JA, and ET production at the infection site during root feeding [87]. Mobile nematodes, on the other hand, cause tissue damage due to intracellular migration. This may lead to damage-triggered immune responses before establishing permanent feeding sites [64].

Many studies have focussed on plant-mediated interactions between herbivores and phytopathogens, yet the underlying mechanisms are not entirely clear. We have general predictions on tripartite effects based on a trophic strategy of pathogens, but the role of phytohormones in this context is uncertain due to varying and contradictory outcomes. Lazebnik et al. [83] proposed three hypotheses. The first states that biotrophic pathogens facilitate chewing herbivores through SA induction and JA downregulation unless the plant exhibits effector-triggered immunity (ETI). Second, biotrophs can either facilitate or inhibit phloem feeders. This is because different pathogens with the same trophic strategy have been found to induce varying effects on aphids (Hemiptera: Aphididae). Aphids are phloem-feeders and can trigger JA/ET as well as SA signalling. It is thought that as aphids can alter plant cytokinin levels and thus source-sink nutrient flows, this facilitates host colonization as well. The third hypothesis states that necrotrophs can inhibit both sucking and chewing insects, as the mechanisms around plant-mediated effects of necrotrophs on insects are not defined yet. Inhibition can be defined as plant-mediated competition and has been found to occur within and across different antagonist species.

### 3.2. Competition

Contrary to facilitative interactions among antagonists, there is also the potential for competition. For instance, insects that induce similar hormonal pathways are more likely to compete [88]. The induction of a hormone involved in plant defences common to two herbivores will, therefore, likely induce defence responses effective against both herbivores. Similar observations have been made when different taxa infest hosts if the same hormonal pathway is induced. Phloem feeders are thought to induce plant resistance to hemibiotrophic and biotrophic bacterial and fungal pathogens due to SA-mediated responses. Chewing insects, on the other hand, can either facilitate or inhibit (hemi)biotrophs. Only a few studies have researched the effects on necrotrophic pathogens post herbivory, in most of which there were no effects [83].

A meta-analysis conducted by Moreira et al. [89] showed that the overall pattern across bioassays involving tripartite interactions between plants, herbivores, and pathogens did not line up with the hormonal antagonism hypothesis of Lazebnik, Frago [83]. Instead, JA-inducing antagonists had a significant negative effect on both subsequent JA- and SA-inducing antagonists. Conversely, SA-inducing antagonists had no significant effect on attackers triggering either of the hormone signalling pathways. Notably, their study proved that the type of species of the initial plant antagonist plays an important role in predicting the plant-mediated effects. JA-triggering insects significantly affected subsequent herbivores associated with both pathways, while SA-triggering insects did not for either. Upon initial infection by pathogens, on average, there was no effect on SA- or JA-triggering herbivores. Furthermore, there appears to be a stronger interactive effect when JA-initial-herbivores are present on the same plant part as the subsequent insect herbivores. In general, there was a higher variability in the effects of SA antagonists than JA antagonists. According to the authors, this is due to the greater diversity of plant responses that can be triggered by the species in the SA-inducing group. Furthermore, they hypothesized that the observations of their study can be explained by several factors. The first is that the studied plants were model species such as *Arabidopsis thaliana*, tobacco, or tomato, which are not representative of plants overall. Second, considering that plant responses to biotic stresses are highly species-specific, the available literature may still be lacking in species diversity, and therefore general conclusions cannot yet be made. The final factor is that certain variables that are specific yet important, such as whether plants are inoculated aboveground or belowground, differ in these studies [89].

Regarding localization, the review paper by Biere and Goverse [64] describes the most notable findings made in cross-compartment studies on resistance. (Hemi)biotrophs can induce SAR across plant tissue compartments, for example, by infection of shoots and subsequent expression in roots. Additionally, root infection by (hemi)biotrophic fungi strongly enhances SA in leaves. Root infection with soilborne fungi generally leads to large transcriptional changes in shoots, which has consequences for aboveground pathogens (SAR). The effects of foliar pathogens on soilborne herbivores are poorly known, neither are the effects of soilborne pathogens on foliar herbivores. Conversely, many studies have investigated the cross-compartment effects of insect attacks and demonstrated that foliar herbivory leads to the induction of transcriptional changes in roots. Phloem-feeders such as aphids and whiteflies (Hemiptera: Aleyrodidae) can induce both SA and JA/ET pathways in host roots. Additionally, their herbivory has mixed effects on belowground (hemi)biotrophs and chewing herbivores. Leaf-chewing insects induce ET responses in roots and enhance plant resistance to belowground herbivores while affecting resistance to soil-borne pathogens. Similarly to aboveground herbivory, insect-caused root damage leads to large transcriptional changes in shoots linked to JA/ET induction. Belowground chewers generally facilitate phloem-feeding insects, as the nutritional status (i.e., amino acids and nitrogen) of the host plant has been enhanced. Conversely, the effects on leaf chewers are negative due to increases in ABA-mediated or other defences such as terpenoids, pyrrolizidine alkaloids, and glucosinolates. Belowground herbivory can enhance resistance to biotrophic and necrotrophic pathogens [64].

There are similar reports that belowground nematodes can affect other antagonists and vice versa. Migratory and sedentary endoparasitic nematodes feeding on roots induce major systemic alterations in monocot and dicot shoots, including in gene expression and defence responses. For instance, tissue damage can lead to the accumulation of PR proteins, POX, and catalase (CAT) [90]. Additionally, it can trigger auxin accumulation and *de novo* root organogenesis through JA-dependent pathways [91]. Furthermore, nematode root feeding leads to altered resource allocation [92], affecting the performance of shoot-feeding insects. Resistance to aboveground phloem-feeding insects is generally enhanced by root nematodes, while the effects on leaf-chewing herbivores depend on the life strategy of the nematode [64]. It was found that root invasion by the root knot nematode (RKN) *Meloidogyne incognita* indirectly affected foliar oviposition by the two-spotted spider mite *Tetranychus urticae* and vice versa. Both antagonists showed a preference for plants that were not already attacked by the other [93]. Few studies have investigated cross-compartment interactions between nematodes and foliar pathogens. In general, plant susceptibility to foliar pathogens is increased by the presence of nematodes, but the effects of aboveground pathogens on nematodes are poorly known. Phloem feeders have been found to have negative effects on PPNs, whereas leaf-chewing insects are reported to increase host susceptibility to nematodes [64]. Even though most nematodes live in soil, some can infect plants foliarly by moving on water films in shoots and subsequently living as ectoparasites that feed off leaves and seeds [94]. Fungal infection can be facilitated due to tissue damage, or, alternatively, the induced systemic responses may enhance plant resistance to other plant antagonists.

## 4. Silicon as a Mediator of Plant Antagonist Interactions

Antagonists can thus either facilitate or compete via their shared host plants, potentially mediated by the phytohormonal pathways they trigger. However, the precise mechanisms through which Si enhances plant resistance against pests and diseases remain speculative in part, as the extent to which hormones are always involved in these mechanisms is uncertain. The question, therefore, is what the effects are of Si supplementation on a host that is under attack by multiple antagonists. Will the plant be less susceptible to all antagonists, or will there be competition due to hormonal crosstalk? There is limited literature on the relationship between Si and interactions between plant antagonists, especially in the context of hormonal signalling pathways, but some synthesis is possible.

Si defences have been linked to phytohormonal responses such as JA levels in rice or *Brachypodium distachyon*, albeit with contrasting patterns [10,40,42,43,95]. A study by Biru et al. [88] demonstrated that prior feeding by caterpillars (*H. armigera*) reduced cricket (*Acheta domesticus*) performance via Si-mediated responses in *B. distachyon*. Considering that Si deposition persists over a relatively long time scale [96], with Si defences once induced taking over a year to decline to the levels in undamaged tissues [97], subsequent interspecific herbivore competition may last longer than competitive interactions mediated by secondary metabolites, where relaxation of these defences is more typically measured in days or weeks [98,99,100]. Additionally, Si-based defences are rapidly induced upon absorption by plants [33] and can be directed to sites of damage [101]. In a study conducted by Islam et al. [82] the impacts of Si supplementation on the contemporaneous performance and interguild interactions between a chewing and a sucking insect, *H. armigera* and *Rhopalosiphum padi*, respectively, were assessed. It was found that the relative growth rate (RGR) of *H. armigera* caterpillars was negatively correlated with *R. padi* aphid abundance on shared host plants. Additionally, reduced caterpillar RGR on Si-amended plants benefitted aphid colonization. The results provide evidence for plant-mediated effects of Si on interspecific competition between two insect herbivores [82]. Considering that these two species induce different phytohormone signalling pathways, these observations suggest that JA and SA were involved in the Si-mediated effects (Figure 2).

The effects of prior herbivory on the performance of subsequent herbivores are highly variable. For example, Johnson et al. [35] found that *R. padi* aphid herbivory induced SA, and the chewing herbivore *H. armigera* triggered JA in *B. distachyon*. Additionally, the chewing herbivores induced Si uptake, and their growth rates were reduced by 75% when feeding on Si-amended plants. However, even though aphids suppressed JA levels, the performance of *H. armigera* subsequently feeding on the same plants remained suppressed on Si-supplemented plants: Si defences were operating regardless of the SA-JA crosstalk [35]. Nevertheless, several studies have observed that Si-mediated defence responses are positively linked to JA. Triggering of the JA pathway has been shown to boost Si defences in rice [42] and *B. distachyon* [35]. Rice was inoculated with the chewing herbivore rice leaffolder (*Cnaphalocrocis medinalis*), which heightened JA accumulation levels in Si-pretreated plants upon attack. Additionally, the application of authentic herbivory or chemical induction with methyl jasmonate (MeJA) Si stimulated the activity of defence-related enzymes such as PPO and POX [42]. In contrast, another study reported that Si supplementation in rice significantly reduced JA production under wounding stress [43]. However, antioxidants were reduced upon Si application, and similarly to Ye et al. [42]’s findings, CAT, POX, and PPO activities were heightened in Si-treated and wounded plants. Vivancos et al. [102] concluded that Si-mediated resistance in *A. thaliana* against powdery mildew (*Golovinomyces cichoracearum*) is conferred through mechanisms other than the SA pathway. Si-supplemented *A. thaliana* plants, genetically modified to contain the Si-transporter TaLsi1 from wheat to enable Si uptake, showed higher resistance to powdery mildew compared to Si-deficient plants. Additionally, Si treatment led to priming via SA accumulation. However, SA-deficient mutants (*pad4* and *sid2*) displayed similar phenotypes, indicating that other mechanisms, such as effector interference, are at play in Si-mediated resistance [102].

Herbivory induces Si accumulation to a greater extent in response to herbivore attack than mechanical damage but may require a threshold level of damage for induction to occur: repeated damage of grass leaves by locusts increased foliar Si concentrations more than single damage events or cutting by scissors [103]. Such induction of Si levels potentially diminishes feeding and/or performance of subsequently attacking insects on these plants [88]. Interactions across taxa mediated by Si induction have also been reported. Si concentrations in the foliage of tall fescue (*Festuca arundinacea*) have been shown to increase due to colonization by symbiotic *Epichloë* spp. endophytic fungi [104,105], which could impact herbivores. The interaction between *Epichloë*-infected grasses, Si-supplementation, and *H. armigera* demonstrated that endophyte-mediated responses and Si-based defences can work complementarily to create resistance to herbivory. Although Si supply did not interfere with *Epichloë*-induced alkaloid production, the immune response of *H. armigera* was reduced through decreased melanisation [106]. Notably, *Epichloë* is not a plant antagonist but a plant-beneficial mutualist. To date, there is little information on the relation between Si and simultaneous cross-species infestations by plant pathogenic fungi or plant parasitic nematodes. Most studies have focussed on herbivorous insects, highlighting a clear knowledge gap in this field.

## 5. Conclusions and Future Prospects

Si-mediated interactions between plant antagonists are potentially more widespread and consequential than those mediated by changes in plant nutrition and secondary metabolites for at least three reasons. Firstly, unlike secondary metabolite-based plant defences, which can be targeted at specific taxa, Si defences are effective against extremely diverse plant antagonists. The prospect for Si mediating antagonist interactions is therefore potentially greater than mechanisms that involve changes to primary and secondary chemistry. Secondly, the induction and deposition of Si defences generally persist over long time timescales, unlike changes in nutrition or many secondary metabolites. Hence, interactions mediated by Si defences are likely to be more persistent. Finally, the ubiquity and reliance of grasses on Si for resistance to biotic antagonists suggests that this could be an important driver of community composition in many grassland and agricultural ecosystems.

However, there is a limited number of studies focussed on Si-mediated plant antagonist interactions. Areas in which our understanding can still be improved are, for instance, the focus on plant and antagonist species, biochemistry, and field testing. While there are many gaps in our current knowledge, these can be addressed. For example, manipulating Si defences via supplementation is experimentally tractable. Additionally, we have ample information about how Si defences operate against individual attackers, which makes it easier to develop evidenced-based hypotheses for how these changes might affect other contemporaneous plant antagonists. If Si supplementation is to be used in future crop protection strategies, as suggested by Guntzer, Keller [107] and Kelland, Wade [108], we propose that we must fully understand whether this resistance persists or is compromised when the plant is under attack by multiple antagonists. We intend this mini review to stimulate further interest in this area.

## Figures and Tables

**Figure 1 plants-14-01204-f001:**
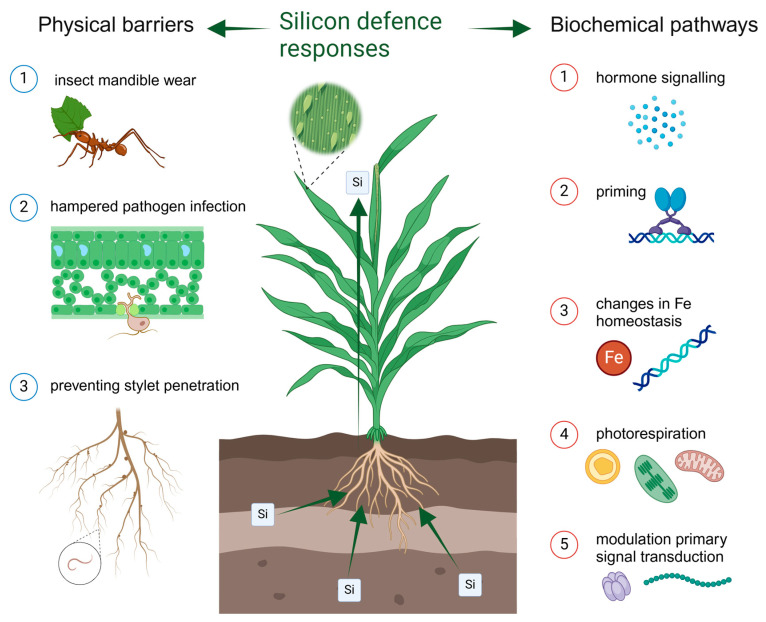
Si-based plant defence is primarily comprised of Si acting as a physical barrier in leaf tissues, but evidence indicates that Si may be involved in biochemical pathways, leading to increased resistance as well. Plants take up Si with their roots from soil, which leads to accumulation of insoluble silica in leaves. Van Bockhaven et al. [5] proposed five hypotheses of plant physiological and biochemical mechanisms leading to Si-induced broad-spectrum resistance.

**Figure 2 plants-14-01204-f002:**
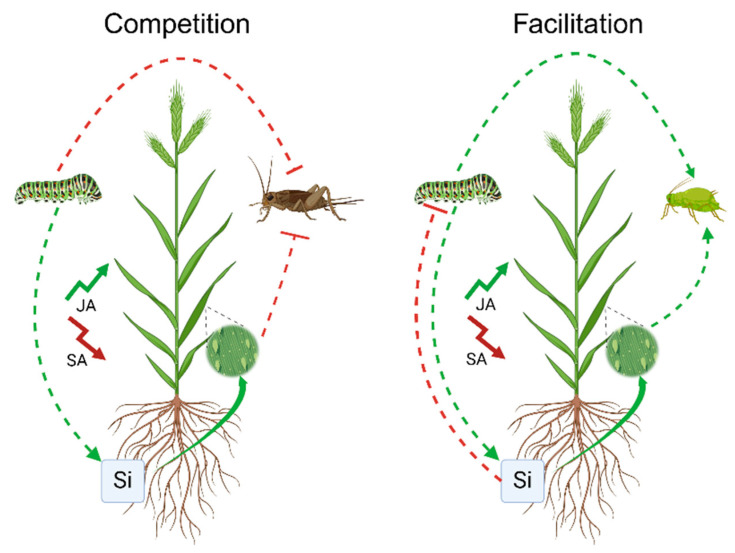
Plant-mediated antagonist interactions influenced by Si uptake, supporting the antagonistic crosstalk hypothesis. Based on findings from Biru et al. [88] and Islamet al. [82]. Green arrows indicate induction and facilitation, while red arrows indicate suppression and competition.

## Data Availability

No new data were created or analyzed in this study.

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
