# Peer review of "Silicon-Mediated Interactions Between Plant Antagonists"

_plants, 2025, doi:10.3390/plants14081204_

Round 1
Reviewer 1 Report
Comments and Suggestions for Authors
Review report
In general, the review article of Denarié et al. would be of interest to the readers of Plants from my point of view. However, I identified several shortcomings that should be addressed before potential publication (please see my detailed comments to the authors below).
- Please avoid teleological wording in your manuscript (e.g., ll.11, 14, 386, 387, 404, 433/434). As far as we know today, evolutionary processes are running in a passive way. Thus, grasses do not use Si to defend themselves against antagonists, but grass species that have higher silica accumulation rates might be more successful in reproduction as they are better protected against attacks.
- Please introduce abbreviations also in the abstract (JA and SA).
- 22. Please correct: not been found not to be…
- 50/51. There are studies that doubt active transport of silicic acid (see, e.g., Exley et al. 2020, Silicon 12:2641-2645). This aspect should be considered, especially in a critical review article.
- 84/85. There are studies that question the wearing of herbivore mouthparts (see, e.g., Kvedaras et al. 2009, Agricultural and Forest Entomology 11, 301-306). Again, a critical review article should always consider two sides of the same coin.
- 89. Correct to “piercing-sucking insects”.
- 93-98. What about the apoplastic obstruction hypothesis by Coskun et al. (2018, New Phytologist)? By reviewing the existing literature, these authors found no distinct evidence that Si affects a wide range of molecular-genetic, biochemical or physiological processes. In this context, see also Exley 2015 (Frontiers in Plant Science 6:853).
- Please be consistent with the use of “Si” or “Silicon” at the beginning of a sentence in your manuscript.
- 104/105. What about foliar applied Si fertilizers/sprays? They have also been found promising in crop protection, especially against fungal infections (see, e.g., Puppe and Sommer 2018, Advances in Agronomy 152).
- 112-125. Actually, there have already been studies in the early 1990s that hypothesized other mechanisms (see, e.g., Chérif et al. 1992, Physiological and Molecular Plant Pathology 41, 371-385).
- 116-125. Why have you only chosen the review article of Van Bockhaven et al. (2013) as source for hypotheses? What about the apoplastic obstruction hypothesis (see my comment no. 7)? For a critical review article, it is mandatory to describe all perspectives on a topic.
- You introduced a lot of abbreviations. Please check if all of them are really needed. If an abbreviation occurs only one or two times in the main text, it should be avoided from my point of view (it is hard for the reader to follow a text with too many abbreviations).
- 139-142. Actually, it does. The review article of Coskun et al. (2018) unifies the current state of knowledge in one model/hypothesis. Please carefully check the review article of Coskun et al. (2018) again, especially their Fig. 5.
- 154. Correct to “cell walls”.
- 1. This figure should be reworked based on my comments nos. 7 and 13.
- Subsection 3.1. Again, please avoid teleological writing. In this subsection it reads like there is a “joint combat plan” of antagonists to attack plants. I would recommend using phrases like “benefit from pre-infections” rather than “can facilitate”, for example.
- 276-278. This should be stated then already in subsection 3.1, where you refer to this article and the corresponding hypotheses.
- 307-311. Is it meaningful to give such general statements here as most studies used only a few (model) plants as you mentioned before (ll.291-293)?
- 2. I really miss a legend or written explanation in the caption for this figure (e.g., what is meant with the arrows etc.).
- 2. As far as I can see, there is no reference to Fig. 2 in subsections 3.1. and 3.2.
- 387. Is there really an active transportation to the sites of infection or is silica precipitating at these sites due to higher transpiration? See, e.g., Chérif et al. 1992 (Physiological and Molecular Plant Pathology 41, 371-385), Exley 2015 (Frontiers in Plant Science 6:853), and Exley et al. 2020 (Silicon 12:2641-2645).
- 422. What kind of mechanisms? Please specify.
- As you relate your work also to agriculture, I really miss information on Si bioavailability in agricultural soils. As Si bioavailability is strongly related to Si pools in soils, water supply, and Si fertilization, these aspects should be named at least shortly in your manuscript.
- Future prospects: As you mainly summarize knowledge that has been already summarized in previous review articles (which you are citing), the “future prospects” section could/should be largely improved. In this context, you should work out existing knowledge gaps and how to fill them in a separate subsection from my point of view.
Author Response
- Please avoid teleological wording in your manuscript (e.g., ll.11, 14, 386, 387, 404, 433/434). As far as we know today, evolutionary processes are running in a passive way. Thus, grasses do not use Si to defend themselves against antagonists, but grass species that have higher silica accumulation rates might be more successful in reproduction as they are better protected against attacks.
RESPONSE: Thank you, we changed a few of those that were indeed more suitable in a different type of phrasing (line 11 & 14)
- Please introduce abbreviations also in the abstract (JA and SA).
RESPONSE: Yes, we have adjusted this now in the abstract (line 16)
- 22. Please correct: not been found not to be…
RESPONSE: Thank you, it has been changed now to ‘’…has not been found to be…’’ (line 22)
- 50/51. There are studies that doubt active transport of silicic acid (see, e.g., Exley et al. 2020, Silicon 12:2641-2645). This aspect should be considered, especially in a critical review article.
RESPONSE: We acknowledge that some authors believe that Si uptake is purely through passive transport, but it has now been widely accepted that active transport is involved and this has been demonstrated in many studies, e.g. McLarnon et al. (2017).
- 84/85. There are studies that question the wearing of herbivore mouthparts (see, e.g., Kvedaras et al. 2009, Agricultural and Forest Entomology 11, 301-306). Again, a critical review article should always consider two sides of the same coin.
RESPONSE: We have added this reference to that discussion now for balance (line 87-88).
- 89. Correct to “piercing-sucking insects”.
RESPONSE: This has now been swapped (line 92).
- 93-98. What about the apoplastic obstruction hypothesis by Coskun et al. (2018, New Phytologist)? By reviewing the existing literature, these authors found no distinct evidence that Si affects a wide range of molecular-genetic, biochemical or physiological processes. In this context, see also Exley 2015 (Frontiers in Plant Science 6:853).
RESPONSE: We do discuss Coskun’s apoplastic obstruction hypothesis in the paper (line 139).
- Please be consistent with the use of “Si” or “Silicon” at the beginning of a sentence in your manuscript.
RESPONSE: Thank you, this has been changed to Si (line 16).
- 104/105. What about foliar applied Si fertilizers/sprays? They have also been found promising in crop protection, especially against fungal infections (see, e.g., Puppe and Sommer 2018, Advances in Agronomy 152).
RESPONSE: Yes indeed, this has been added now (line 111-112).
- 112-125. Actually, there have already been studies in the early 1990s that hypothesized other mechanisms (see, e.g., Chérif et al. 1992, Physiological and Molecular Plant Pathology 41, 371-385).
RESPONSE: Indeed, we have altered this accordingly (line 118-119).
- 116-125. Why have you only chosen the review article of Van Bockhaven et al. (2013) as source for hypotheses? What about the apoplastic obstruction hypothesis (see my comment no. 7)? For a critical review article, it is mandatory to describe all perspectives on a topic.
RESPONSE: See previous response and text (line 139-144).
- You introduced a lot of abbreviations. Please check if all of them are really needed. If an abbreviation occurs only one or two times in the main text, it should be avoided from my point of view (it is hard for the reader to follow a text with too many abbreviations).
RESPONSE: This is a good point, we have added a Glossary to facilitate this (line 479-500).
- 139-142. Actually, it does. The review article of Coskun et al. (2018) unifies the current state of knowledge in one model/hypothesis. Please carefully check the review article of Coskun et al. (2018) again, especially their Fig. 5.
RESPONSE: Thank you, we have removed this incorrect statement from the paper.
- 154. Correct to “cell walls”.
RESPONSE: Thank you, it has been changed (line 155).
- 1. This figure should be reworked based on my comments nos. 7 and 13.
RESPONSE: As we say in our responses to comments nos. 7 and 13, we have discussed the Coskun theory in the text but this figure is specifically referring to the 5 hypotheses in Van Bockhaven et al. (2013).
- Subsection 3.1. Again, please avoid teleological writing. In this subsection it reads like there is a “joint combat plan” of antagonists to attack plants. I would recommend using phrases like “benefit from pre-infections” rather than “can facilitate”, for example.
RESPONSE: We acknowledge that teleological writing should be avoided, however ‘’facilitation’’ is a known and accepted term and the usage of it in similar contexts is common too.
- 276-278. This should be stated then already in subsection 3.1, where you refer to this article and the corresponding hypotheses.
RESPONSE: Thank you for this suggestion, however we think that this is more suitably placed in subsection 3.2.
- 307-311. Is it meaningful to give such general statements here as most studies used only a few (model) plants as you mentioned before (ll.291-293)?
RESPONSE: We mention that most studies used a few plant species, so we believe it is also implied that the recorded observations discussed by the referred papers are based on that.
- 2. I really miss a legend or written explanation in the caption for this figure (e.g., what is meant with the arrows etc.).
RESPONSE: We have now added written explanations of the coloured arrows in the caption of the figure 4 (line 398 and 399).
- 2. As far as I can see, there is no reference to Fig. 2 in subsections 3.1. and 3.2.
RESPONSE: Yes, we refer to Fig.2 in section 4 (line 375) so we have moved it now to a more suitable spot (line 376-399).
- 387. Is there really an active transportation to the sites of infection or is silica precipitating at these sites due to higher transpiration? See, e.g., Chérif et al. 1992 (Physiological and Molecular Plant Pathology 41, 371-385), Exley 2015 (Frontiers in Plant Science 6:853), and Exley et al. 2020 (Silicon 12:2641-2645).
RESPONSE: Based on numerous studies showing herbivore induced silicon uptake, we can conclude that it cannot be achieved via transpiration alone (McLarnon et al 2017). Furthermore, multiple silicon transporters have been found, including active proton pumps (Ma & Yamaji, 2006). The paper of Deshmukh & Bélanger (2016) suggests that plants need to have active transport to get to more than 1% dry mass. Furthermore, the Thorne et al 2023 paper showed the redirection of Si towards damage sites is through mobilisation of soluble Si in the phloem, which is clearly not driven by transpiration.
- 422. What kind of mechanisms? Please specify.
RESPONSE: This is a good point, we have added the mechanism that is suggested in Vivancos’s paper, i.e. effector interference (line 425).
- As you relate your work also to agriculture, I really miss information on Si bioavailability in agricultural soils. As Si bioavailability is strongly related to Si pools in soils, water supply, and Si fertilization, these aspects should be named at least shortly in your manuscript.
RESPONSE: We agree with the point that is being made, therefore we have added this to the text (line 52-54).
- Future prospects: As you mainly summarize knowledge that has been already summarized in previous review articles (which you are citing), the “future prospects” section could/should be largely improved. In this context, you should work out existing knowledge gaps and how to fill them in a separate subsection from my point of view.
RESPONSE: Thank you for the suggestion. We have added some of the existing knowledge gaps to the future prospects paragraph (line 460 and 461).
Reviewer 2 Report
Comments and Suggestions for Authors
Silicon accumulation is a strategy used mainly by grasses providing a physical barrier against antagonists. Biochemical pathways may also be involved in Si- mediated plant resistance, although the precise mode-of-action in this case is less clear. In this review, the authors focus on consider how plants deploy Si resistance when simultaneously and/or sequentially attacked by insect herbivores, fungal phytopathogens, and plant parasitic nematodes, and how the plant hormones JA and SA are involved. In general, this manuscript was well-written and summarized the recent progress the effect of Si on plant-mediated interactions between insect herbivores within the same feeding guild and across different feeding guilds and how the plant hormones JA and SA are involved. I have the following comments:
- The review paper has only two figures and no table, it seems the content not deep enough. More figure or Table should be added.
- As the review paper, there is no referenced cited from 2024 and 2025, which is not acceptable. Thus, latest references should be added in the reversion.
- The first letter of the key word should be capital and key words should be separated by semicolon.
Author Response
- The review paper has only two figures and no table, it seems the content not deep enough. More figure or Table should be added.
RESPONSE: Thank you, however we elaborately discuss the current knowledge regarding the subject of our paper. Considering that the existing observations are limited, the figures in our manuscript encompass all of it in a succinct manner.
- As the review paper, there is no referenced cited from 2024 and 2025, which is not acceptable. Thus, latest references should be added in the reversion.
RESPONSE: Actually we cited several research papers from 2024, such as Lata-Tenasaca et al. (2024) (line 138) and Qi et al. (2024) (line 149) and we have now added some from 2025: Do Prado Mattos et al. 2025 (line 105) and Leal et al. (2025) (line 112).
- The first letter of the key word should be capital and key words should be separated by semicolon.
RESPONSE: Thank you, we have changed this as recommended (line 25).
Round 2
Reviewer 1 Report
Comments and Suggestions for Authors
The authors revised their MS based on my critical comments. Although I would have preferred a more critical writing regarding some aspects in this MS, I respect the authors' perspective and interpretation of existing literature, of course.
Thus, I recommend to accept the revised version of this MS. Congratulations.
Reviewer 2 Report
Comments and Suggestions for Authors
Since the authors addressed all my quetions, thus, I have no further comments.